# Are the Eatwell Guide and Nutrient Profiling Models Consistent in the UK?

**DOI:** 10.3390/nu13082732

**Published:** 2021-08-09

**Authors:** Ana-Catarina Pinho-Gomes, Asha Kaur, Peter Scarborough, Mike Rayner

**Affiliations:** 1Faculty of Life Sciences & Medicine, School of Population Health & Environmental Sciences, King’s College London, London WC2R 2LS, UK; cat.pinho-gomes@kcl.ac.uk; 2The George Institute for Global Health, London W12 0BZ, UK; 3Nuffield Department of Population Health, University of Oxford, Oxford OX3 7LF, UK; asha.kaur@ndph.ox.ac.uk (A.K.); peter.scarborough@ndph.ox.ac.uk (P.S.)

**Keywords:** nutrient profiling model, food-based dietary guidelines, food policy, diet

## Abstract

A nutrient profiling model (NPM) was developed in 2005 in the UK to regulate the marketing of foods to children. It was revised in 2018, but the new version has not been finalised. The Eatwell Guide (EWG) is the UK’s official food-based dietary guidelines. The aim of this study was to evaluate the agreement between the 2005 and 2018 versions of the NPM and the EWG. Using recent National Diet and Nutrition Surveys, we estimated the healthiness of individual diets based on an EWG dietary score and a NPM dietary index. We then compared the percentage of agreement and Cohen’s kappa for each combination of the EWG score and NPM index across the range of observed values for the 2005 and 2018 versions. A total of 3028 individual diets were assessed. Individuals with a higher (i.e., healthier) EWG score consumed a diet with, on average, a lower (i.e., healthier) NPM index both for the 2005 and 2018 versions. Overall, there was good agreement between the EWG score and the NPM dietary index at assessing the healthiness of representative diets of the UK population, when a low cut-off for the NPM dietary index was used, irrespective of the version. This suggests that dietary advice to the public is broadly aligned with NPM-based food policies and vice-versa.

## 1. Introduction

Nutrient profiling is the “science of classifying or ranking foods according to their nutritional composition for reasons related to preventing disease and promoting health”, and nutrient profiling models are algorithms that classify or rank foods for the purposes of preventing disease and promoting health [1]. The nutrient profiling model (NPM) in the UK was developed in 2005 by the Food Standards Agency to regulate marketing of foods to children [2]. The model uses a simple scoring system that recognises the benefits of a balanced nutritional diet by awarding negative points to components that children should eat more (i.e., protein, fibre, fruit and vegetables, and nuts), and positive points to foods with components that children should reduce in their diet (i.e., energy, saturated fats, sodium, and sugars). A final score is calculated as the total of positive and negative points, which means a lower score indicates a healthier food. Foods and drinks that score above four and one, respectively, face marketing restrictions [2].

The NPM was reviewed and modified by Public Health England in 2018. This draft was open for public consultation and published, but it was never finalised [3]. This version of the NPM was updated to incorporate new evidence on the association between nutrient intakes and health outcomes, such as systematic reviews linking the consumption of certain food groups with health outcomes [4,5,6,7]. The key changes were a reduction in total energy used as reference from 2130 kcal to 2000 kcal, a change from total to free sugars, and a reduction in the recommended intake of sugars (21% of food energy from total sugars versus 5% from free sugars), as well as an increase in fibre from 24 g to 30 g per day. The points attributed to foods according to content in energy, saturated fat, free sugars, and fibre were adjusted accordingly.

It is generally agreed that NPMs should rank and classify foods in ways that are consistent with food-based dietary guidelines (FBDGs) [8]. However, the extent to which NPMs complement and/or are consistent with FBDGs has been a source of controversy for some time. This is, at least partly, due to the lack of a consensual method for testing the agreement between NPMs and FBDGs, even though several methods have been proposed over the years [9,10].

The Eatwell Guide (EWG) constitutes the UK’s official FBDGs, and was updated in 2016 using optimisation modelling [11]. The EWG is based on a set of recommendations that the population should follow to eat a healthy diet. These include both nutrient- and food-based recommendations, which are converted into advice on how much to eat from each food group to achieve a healthy, balanced diet. It was developed in a way that minimised the dietary changes required for the population to achieve the recommended levels of consumption of each food group in view of the baseline levels of consumption.

It is critical that the NPM and EWG are aligned to ensure that public health interventions that are underpinned by the NPM, such as the regulation of broadcast advertising of foods and volume-based promotions [12], are consistent with the government’s recommendations for healthy eating, which are used by consumers, food manufacturers, and retailers. As marketing is a key determinant of food behaviour [13], particularly in children [14], NPMs should support the recommendations of FBDGs. However, the NPM and EWG were developed independently and operate at different levels (on foods and diets, respectively). Therefore, it is uncertain whether food classifications from the NPM, when aggregated at diet level, produce classifications that are in tune with the EWG. It is also uncertain whether the updated version of 2018 has had any impact on the alignment between the NPM and the EWG. Therefore, the aim of this study was to evaluate to what extent the NPM was consistent with the EWG, comparing the 2005 and the 2018 versions.

## 2. Methods

### 2.1. Data Source

This study used individual food data from years 9 to 11 of the National Diet and Nutrition Survey (NDNS) for the years 2016/2017 to 2018/2019 [15]. Children up to 5 years of age were excluded from the NDNS data, as the EWG recommendations are not applicable to this age group [16]. The NDNS rolling programme is a continuous, cross-sectional survey, which is designed to collect detailed, quantitative information on the food consumption, nutrient intake, and nutritional status of the general population aged 1.5 years and over living in private households in the UK. The survey covers a representative sample of around 1000 people per year. The first stage comprises a face-to-face computer-assisted personal interview with each participant, completion of an estimated four-day food diary by the participant, measurements of height and weight, and collection of a spot urine sample (for those aged 4 years and over). Participants who take part in the interview and complete a food diary for at least 3 days are invited to take part in the second stage of the survey, which involves a visit from a nurse to take further physical measurements and a blood sample. A detailed description of the methods underpinning the NDNS is published elsewhere [17].

### 2.2. Data Analysis

#### 2.2.1. Healthiness of Diet Based on Adherence to EWG

Dietary intakes reported in the NDNS were compared with recommended intakes that underpin the EWG (Appendix A). Participants were given a point (1) if they met at least the minimum daily intake for recommended foods and nutrients and (2) if they did not exceed the maximum daily intake for food and nutrients that was recommended. Participants’ diets were assessed against each of the recommendations using the average intake per day (e.g., average intake of fruit and vegetables per day for the days available). The recommendation for energy intake was not included because daily energy requirements are highly variable between individuals.

Participants were grouped into two categories of adherence based on the number of recommendations met (total = 10): low adherence or less healthy diet (score 0 to 5) versus high adherence or healthier diet (score 6 to 10).

#### 2.2.2. Healthiness of Diet According to the NPM 2005 and 2018

First, for each food or beverage in the NDNS food composition database, we calculated the NPM 2005 and 2018 score (food-level score) based on its composition for each 100 g of content, using the published scoring systems (Appendix A) [3].

Second, we calculated a NPM dietary index to characterise the nutritional quality of each individual’s diet. The NPM dietary index (individual-level score) was computed as the sum of NPM score for each food or beverage consumed, multiplied by the amount of energy provided by this product (energy content per 100 g multiplied by the estimated daily intake assessed using the baseline dietary questionnaires), divided by the total amount of energy intake [18]. A higher NPM dietary index value reflected an overall lower nutritional quality of foods consumed (i.e., a less healthy diet).

#### 2.2.3. Comparison of EWG and NPM 2005 and 2018

We investigated whether there was an association between the EWG score and the NPM dietary index by (1) calculating the mean NPM dietary index for each of the levels of the EWG score, (2) plotting the distribution of the NPM values by level of EWG score, and (3) calculating the Pearson correlation coefficient. We did this for both versions of the NPM (i.e., 2005 and 2018).

Using the aforementioned binary classification of the diet based on adherence to the recommendations underpinning the EWG (i.e., healthier versus less healthy), we estimated the concordance between the EWG score and the NPM dietary index using different cut-offs for defining healthier versus less healthy diets according to the NPM dietary index. We compared the versions of 2005 and 2018. We calculated the percentage of agreement and Cohen’s kappa for each combination of the EWG score and NPM dietary index across the range of observed values for the 2005 and 2018 versions of the NPM dietary index.

## 3. Results

We included data from years 9 to 11 of the NDNS (2016/2017 to 2018/2019), with a total of 3028 individuals (1062 for year 9, 1025 for year 10, and 941 for year 11). A detailed description of the participants in the NDNS survey and their diets is published elsewhere [17].

Individual diets achieved 0–9 points out of 10 possible points on the EWG score (Table 1). The recommendations that were met by the largest number of individuals were those related to salt, red and processed meat, and protein, whilst recommendations regarding fibre, fish, fruit and vegetables, and free sugars were the least commonly achieved (Table 2). The NPM dietary index varied between −4 and 14 for the 2005 version and between −3 and 17 for the 2018 version. The mean NPM dietary index decreased as the EWG score increased, as a lower NPM dietary index and a higher EWG score reflect a healthier diet (Table 1). For instance, for individuals who met none of the recommendations underpinning the EWG (i.e., those with an EWG score of zero), the mean NPM dietary index was 9.47 (SD 1.98) and 7.81 (SD 2.08) using the 2018 and 2005 versions, respectively. For those who achieved nine of the recommendations underpinning the EWG (i.e., those with a score of nine), the mean NPM dietary index was 1.21 (SD 2.23) and 0.30 (SD 2.47) for the 2018 and 2005 versions, respectively. For each point on the EWG score, the NPM dietary index was, in general, higher for the 2018 version than the 2005 version. This suggests that diets tended to be considered less healthy by the 2018 version than the 2005 version of the NPM dietary index at each level of the EWG score.

Diets that achieved a high EWG score had, on average, a lower NPM dietary index using both the 2005 and 2018 versions (Figure 1 and Figure 2). The distribution of the NPM dietary index stratified by level of the EWG score also showed that diets classified as healthier according to the EWG score had a lower mean NPM dietary index using the versions from 2005 and 2018 (Figure 3 and Figure 4). However, within each level of the EWG score, there was substantial variation in the NPM index achieved by individuals’ diets, as shown by the approximately normal distribution of the NPM index for each level of the EWG score.

There was a low, yet statistically significant, correlation between the EWG score and the NPM dietary index, with no evidence of a difference between the 2005 and 2018 versions (correlation coefficient −0.45 for the NPM version 2005 and −0.43 for the NPM version 2018, Appendix A). This showed that as the EWG score increased, reflecting diets becoming healthier, the NPM dietary index achieved by those diets decreased, which also reflects healthier diets.

Considering that adhering to six or more recommendations underpinning the EWG corresponded to a healthier diet (i.e., meeting at least half of the 10 recommendations), the percentage of agreement between the EWG score and the NPM dietary index was comparable for the versions of 2018 and 2005, when an NPM dietary index threshold to define healthier diets was set at three (Table 3). When the threshold to discriminate between healthier and less healthy diets was set at a higher value, the 2018 version appeared to have better agreement with the EWG score than the 2005 version. However, Cohen’s kappa suggested that agreement between the EWG score and the NPM dietary index was low to moderate, irrespective of the NPM version. Cohen’s kappa was lower than percentage agreement because the latter did not take into account the possibility of an agreement occurring due to chance. This possibility increases when the population is unevenly split between the two groups, such as when the selected NPM threshold score is either very high or very low. The highest kappa values were observed when a cut-off of three to five was used to consider a diet as healthy using the NPM dietary index of 2018, and of two to three when the 2005 version was used. This suggests that concordance between the EWG score and the NPM dietary index is maximal when diets are slightly healthier for the 2018 version rather than the 2005 version.

## 4. Discussion

This study used a score based on the food and nutrient recommendations underpinning the EWG and a dietary index based on the NPM versions of 2005 and 2018 to evaluate the healthiness of representative diets of the UK population. It demonstrated that, overall, diets considered healthy according to the EWG score achieve a lower NPM dietary index score, irrespective of the version used to calculate that index. The mean NPM dietary index was higher for the 2018 version than the 2005 version of the NPM at each level of the EWG score. Overall, agreement between the EWG score and the NPM dietary index in classifying a diet as “healthy” was good when a low cut-off for the NPM dietary index was used, suggesting that the NPM is broadly consistent with the UK’s FBDGs.

Both FBDGs and NPMs are based on the principle that foods can be classified as healthy if their consumption is associated with a reduced risk of disease or improved health and wellbeing [19]. However, the association between diet and health is complex and observational studies are subject to confounding because individuals who eat “healthy” diets tend to engage with other health-promoting behaviours and lifestyles, live in more affluent areas, and have a higher education level [20]. In addition, a diet that reduces the risk of disease, i.e., a healthy diet, depends not only on the individual foods that constitute the diet, but also on the frequency, amount, and combination in which they are eaten [19]. Therefore, it is difficult to compare NPMs, which assess the healthiness of individual foods, and FBDGs, which recommend what to eat to have a healthy diet. To add complexity, FBDGs do not, for most foods, classify them as either healthy or unhealthy, but rather make recommendations about the composition of a healthy diet based on broad food groups, and within these food groups there is a wide variability in nutrient composition.

Nonetheless, for some interventions aimed at improving health in the population, it is necessary to categorise foods according to their contribution to the healthiness of the diet. NPMs can serve as the basis for this categorisation, and it is thus important to assess whether this categorisation is consistent with other government advice on healthy diets. This study found that the healthiness of actual diets (as measured by compliance with the recommendations underpinning the EWG) reflects differences in the proportions of healthy and/or unhealthy foods as assessed by the NPM. Overall, it supported this assumption, because as the NPM dietary index decreased as the EWG score increased, irrespective of which version of the NPM was used. In addition, healthier diets, i.e., diets with a higher EWG score, had a narrower distribution of NPM dietary index values than less healthy diets. This suggests that healthier diets, as assessed by the EWG score, tended to include mostly foods that would be classified as healthy by the NPM, whilst less healthy diets included foods within a broader range of healthiness according to the NPM. However, the sample size for high and low EWG scores was small, and thus, the findings need to be interpreted with caution.

Although there is no gold standard against which to compare NPMs, different methods have been used to validate NPMs, such as comparing NPMs against FBDGs or health outcomes [21,22,23,24]. There is substantial overlap among FBDGs worldwide, as these are based on evidence on the components of a healthy diet that are associated with a reduced risk of nutrition-related diseases [25]. In keeping with this, NPMs should rank foods according to their healthiness, which should similarly reflect a reduced risk of nutrition-related diseases. Prospective cohort studies have shown an association between adherence to the recommendations underpinning the EWG and improved health outcomes [16]. Therefore, showing that the NPM dietary index is broadly concordant with the EWG score suggests that it ranks foods as healthy and less healthy by applying similar criteria to those employed by the EWG, which have been shown to be associated with improved health. Although further studies are required to confirm whether the 2018 version of the NPM is more consistent with the EWG than the 2005 version, there are possible explanations why this might be the case. First, the recommendation regarding fibre was poorly met by those with low EEG scores in general, and the updated version of the NPM increased the daily requirement for fibre from 24 g to 30 g and increased the points afforded to fibre. This meant that high-fibre foods would achieve lower NPM values (as fibre points are deducted), and those foods would be included in greater amounts in diets with higher EWG scores. Second, the more restrictive allowance for free sugars in the 2018 version of the NPM may have increased its alignment with the EWG, as the recommendation related to free sugars was most commonly met by those with high EWG scores.

NPMs have been broadly used for two main purposes: supporting consumers facing food labelling and the regulation of food marketing and advertising. Although the principles and criteria underpinning NPMs were developed for both purposes and are similar, the way in which they are used can be different. Food labelling based on NPMs can assume that foods are distributed along a continuum of relative nutritional quality ranging from healthier to less healthy [26]. These food labelling systems, such as those used in Australia or France, are typically graded systems that rank the nutritional quality of foods across the range of possible NPM values [23,27]. The healthiness of foods is then displayed using a score that is depicted as stars, letters, or colours. NPMs can also be applied not as a continuous, but as a binary measure, that either allows or prohibits marketing of certain foods and drinks, as happens in the UK. This means that its alignment with the EWG varies according to the threshold that defines which foods are unhealthy, and hence, which are subject to marketing restrictions. Due to the way the NPM dietary index was calculated in this study, it was not possible to determine directly the exact value of NPM that should be used as cut-off to classify foods as healthy or less healthy in order to maximise concordance with the EWG.

Although the scientific merit of applying a binary definition of individual foods as “healthy” or “unhealthy” based on NPM has been debated [28], even in countries where NPMs are used as continuous scores for food labelling, pressure has been mounting to adopt objective criteria defining “unhealthy” foods in order to regulate marketing and advertising [29]. Decisions about whether a certain food can or cannot be advertised require a binary definition of “healthy” and “unhealthy” based on a pre-specified cut-off value of the NPM. However, there is no scientific consensus on the existence of a specific nutritional composition threshold that distinguishes between “healthy” and “unhealthy” foods. For the purpose of regulation and taxation, a binary classification may be unavoidable, but this needs to be carefully explained to the population to avoid unintended consequences of determining that foods are “healthy” or “unhealthy” [30].

National FBDGs provide the overarching framework and benchmark for a healthy diet, based on current knowledge of the associations between various dietary components and health outcomes [31]. It is, thus, important to ensure that the EWG, which provides the official advice on healthy eating to food manufacturers, retailers, and consumers, is consistent with the NPM, which is used to regulate marketing of foods in the country. Otherwise, the population will get mixed messages about healthy diet and food, which can exacerbate the ongoing problem of misleading nutritional claims on food labels and adverts [32,33,34]. This study demonstrated that, overall, the EWG score and the NPM dietary index agree on what constitutes a healthy diet, for low values of the NPM index, using either the 2005 or 2018 versions. This suggests that there is good alignment between the NPM and the EWG. It is important to note, though, that the overlap between NPMs and FBDGs can never be perfect. For example, while salmon falls into the recommended food group of fatty fish, its high fat and salt content, particularly for smoked salmon, can render it “unhealthy” according to the NPM. Rather than invalidating NPMs or FBDGs, these discrepancies emphasise the complementarity between the two approaches at food and diet levels and highlight the need for clear guidance to the public so that they understand these nuances when making food and diet choices.

### Strengths and Limitations

This study has several strengths. It used actual diets from individuals living in the UK to compare the EWG score and the NPM dietary index, which are more relevant than hypothetical diets. Actual diets can take greater account of other factors that are unrelated to health that shape diets (such as the palatability of food), rather than modelled diets. In addition, nutritional composition was available for all foods in a standardised format, which enabled computing the NPM and estimating compliance with the recommendations underpinning the EWG with accuracy.

There are also some limitations worth acknowledging. First, the NDNS has a relatively small sample size, which limited the ability to perform a subgroup analysis according to age, sex, or region. Second, the NDNS relies on self-reported food intake, which may be subject to bias. Third, the NPM dietary index was computed as a continuous variable to grade the nutritional quality of individual diets, whilst this particular NPM was designed to be used to classify foods as healthy or less healthy using a cut-off of four for foods and one for drinks. Therefore, the NPM dietary index cannot be directly interpreted or compared with the NPM value that is calculated for individual foods. In addition, although the NPM 2005 and NPM 2018 categorise foods and drinks as “less healthy” using the same threshold, they are not directly comparable as ordinal measures due to changes to the scales used. Fourth, it is possible that agreement between the EWG score and the NPM dietary index varies according to the context and population (e.g., alignment may differ between adults and children). However, those limitations are unlikely to have had a material impact on the key findings of this study. Fifth, salt consumption was based on salt that is included in foods (either naturally occurring or added during processing), but not salt added at the table, which means that individual salt consumption may be underestimated. Sixth, we applied generic food and nutrient recommendations to the entire population instead of sex- and age-specific recommendations. Nonetheless, this had no material impact on the study findings as the purpose was to compare different classification systems, which were applied irrespective of demographic characteristics of the individuals. Age- and sex-specific recommendations would have been important if the aim was to evaluate individual diets.

## 5. Conclusions

In conclusion, this study suggested that a dietary score based on the recommendations underpinning the EWG and a dietary index based on the NPM versions of 2005 and 2018, which are broadly concordant when assessing the healthiness of representative diets of the UK population. NPMs and FBDGs are increasingly used by food manufacturers, retailers, and consumers to make informed decisions about healthy eating. Therefore, NPMs and FBDGs should continue to evolve in parallel in response to new evidence on the impact of food and diet on both human and planetary health. Further research is warranted to understand how to incorporate data regarding the degree of food processing, additives, and environmental considerations (e.g., carbon footprint) into FBDGs and NPMs to allow consumers, manufacturers, retailers, and policy makers to set priorities and make informed decisions that will promote diets that are both healthy and sustainable.

## Figures and Tables

**Figure 1 nutrients-13-02732-f001:**
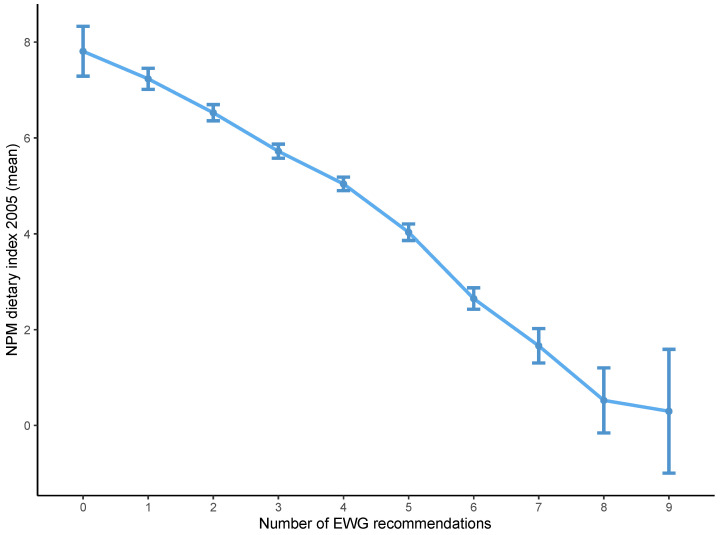
Mean NPM dietary index 2005 according to the number of recommendations underpinning the EWG achieved by individuals. The NPM dietary index was calculated as the weighted average of NPM for all foods included in each individual’s diet. The EWG score was calculated as the total number of recommendations underpinning the EWG that were met by each individual’s diet. The mean of the NPM dietary index was plotted for all individuals achieving each level of the EWG score (0 to 9), with 95% confidence intervals. EWG, Eatwell Guide; NPM, nutrient profiling model.

**Figure 2 nutrients-13-02732-f002:**
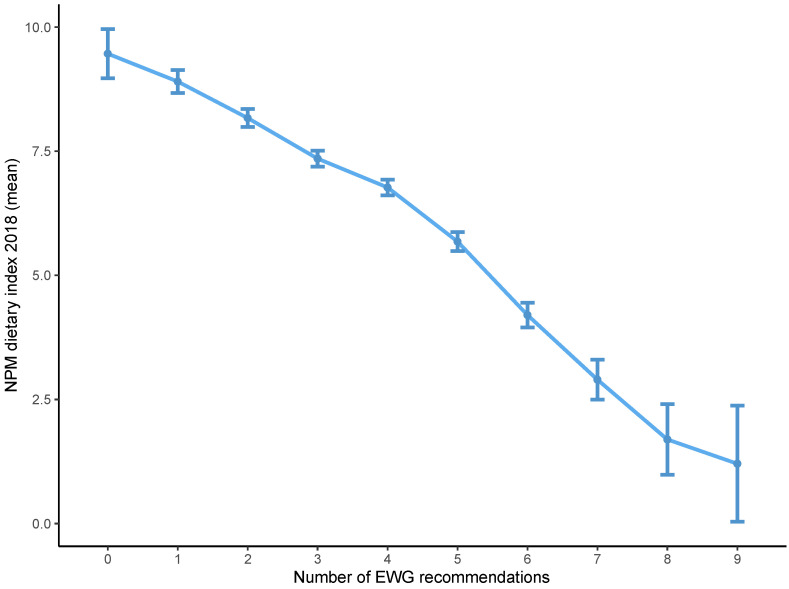
Mean NPM dietary index 2018 according to the number of recommendations underpinning the EWG achieved by individuals. The NPM dietary index was calculated as the weighted average of NPM for all foods included in each individual’s diet. The EWG score was calculated as the total number of recommendations underpinning the EWG that were met by each individual’s diet. The mean of the NPM dietary index was plotted for all individuals achieving each level of the EWG score (0 to 9), with 95% confidence intervals. EWG, Eatwell Guide; NPM, nutrient profiling model.

**Figure 3 nutrients-13-02732-f003:**
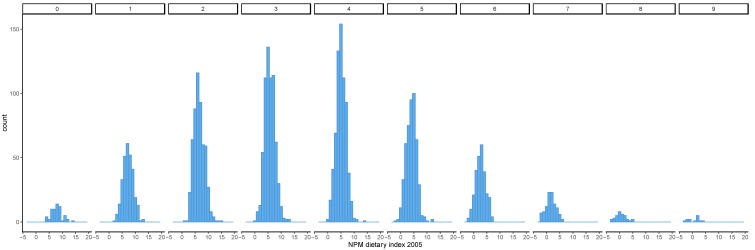
Distribution of the NPM dietary index 2005 by the number of recommendations underpinning the EWG achieved. The NPM dietary index was calculated as the weighted average of NPM for all foods included in each individual’s diet. The EWG score was calculated as the total number of recommendations underpinning the EWG that were met by each individual’s diet. The distribution of the NPM dietary index calculated for individual diets was plotted stratified by level of the EWG score (0–9). EWG, Eatwell Guide; NPM, nutrient profiling model.

**Figure 4 nutrients-13-02732-f004:**
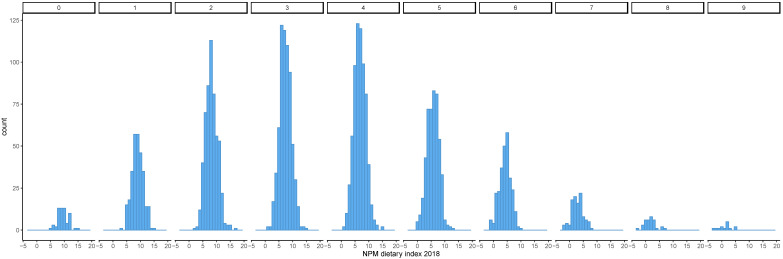
Distribution of the NPM dietary index 2018 by the number of recommendations underpinning the EWG achieved. The NPM dietary index was calculated as the weighted average of NPM for all foods included in each individual’s diet. The EWG score was calculated as the total number of recommendations underpinning the EWG that were met by each individual’s diet. The distribution of the NPM dietary index calculated for individual diets was plotted stratified by level of the EWG score (0– 9). EWG, Eatwell Guide; NPM, nutrient profiling model.

**Table 1 nutrients-13-02732-t001:** Variation of the NPM dietary index calculated using the versions from 2005 and 2018 across levels of the EWG score.

EWG Score	Number Individuals	NPM Dietary Index 2018 (Mean, SD)	NPM Dietary Index 2005 (Mean, SD)
0	61	9.47, 1.98	7.81, 2.08
1	294	8.90, 2.02	7.23, 1.92
2	547	8.17, 2.16	6.53, 2.02
3	661	7.35, 2.11	5.72, 1.94
4	681	6.77, 2.12	5.04, 1.89
5	483	5.68, 2.16	4.03, 1.92
6	269	4.20, 2.09	2.65, 1.87
7	106	2.90, 2.11	1.66, 1.88
8	34	1.69, 2.12	0.52, 2.02
9	14	1.21, 2.23	0.30, 2.47

The NPM dietary index was calculated as the weighted average of NPM for all foods included in each individual’s diet. The EWG score was calculated as the total number of recommendations underpinning the EWG that were met by each individual’s diet. EWG, Eatwell Guide; NPM, nutrient profiling model.

**Table 2 nutrients-13-02732-t002:** Number of individuals meeting each of the 10 recommendations underpinning the EWG.

EWG Score	Protein	Carbohydrates	Free Sugars	Red and Processed Meat	Fish	Fat	Saturated Fat	Fruit and Veg	Fibre	Sodium
**0**	0	0	0	0	0	0	0	0	0	0
**1**	100	11	3	24	7	1	6	3	1	138
**2**	284	69	20	190	32	43	17	29	5	405
**3**	382	220	80	299	75	231	51	74	17	554
**4**	369	381	70	431	90	506	154	103	29	591
**5**	289	329	61	367	77	420	293	108	21	450
**6**	228	189	57	228	69	247	212	109	25	250
**7**	93	78	40	99	43	99	92	80	22	96
**8**	31	27	18	31	22	34	33	32	10	34
**9**	13	10	9	14	13	14	14	14	11	14
**Total**	1789	1314	358	1683	428	1595	872	552	141	2532

The EWG recommendations are detailed in Appendix A. EWG, Eatwell Guide.

**Table 3 nutrients-13-02732-t003:** Agreement regarding the healthiness of diets between the EWG score and the NPM dietary index 2018 and 2005 across different thresholds of the NPM dietary index.

NPM Threshold	NPM Dietary Index 2018			NPM Dietary Index 2005		
	Percentage Agreement	Cohen’s Kappa	*p*-Value	Percentage Agreement	Cohen’s Kappa	*p*-Value
−4	NA	NA	NA	86.6	0.004	0.011
−3	86.6	0.004	0.011	86.6	0.008	<0.001
−2	86.7	0.016	<0.001	86.9	0.040	<0.001
−1	86.9	0.040	<0.001	87.5	0.116	<0.001
0	87.4	0.105	<0.001	88.3	0.216	<0.001
1	88.2	0.216	<0.001	88.9	0.343	<0.001
2	89.0	0.331	<0.001	89.0	0.474	<0.001
3	89.0	0.434	<0.001	86.0	0.481	<0.001
4	87.2	0.461	<0.001	78.2	0.390	<0.001
5	82.7	0.437	<0.001	64.0	0.249	<0.001
6	72.2	0.316	<0.001	49.0	0.148	<0.001
7	59.7	0.219	<0.001	35.5	0.084	<0.001
8	45.6	0.131	<0.001	25.0	0.040	<0.001
9	32.6	0.070	<0.001	19.3	0.019	<0.001
10	24.3	0.037	<0.001	15.8	0.008	<0.001
11	18.6	0.017	0.039	14.3	0.003	0.039
12	15.5	0.006	0.001	13.8	0.001	0.172
13	14.3	0.002	0.048	13.7	0.001	0.297
14	13.8	0.001	0.172	13.5	0.001	0.495
15	13.6	<0.001	0.431	NA	NA	NA
16	13.5	<0.001	0.694	NA	NA	NA
17	13.5	<0.001	0.694	NA	NA	NA

Diets that met six or more recommendations underpinning the EWG were considered as healthy. The NPM dietary index was calculated as the weighted average of NPM for all foods included in each individual’s diet. The EWG score was calculated as the total number of recommendations underpinning the EWG that were met by each individual’s diet. NPM threshold refers to the value of the NPM dietary index used to distinguish between healthier and less healthy diets. EWG, Eatwell Guide; NPM, nutrient profiling model.

## Data Availability

The datasets analysed during the current study are available in the UK data repository available online through https://ukdataservice.ac.uk (accessed on 21 May 2021).

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
