# Peer review of "Are the Eatwell Guide and Nutrient Profiling Models Consistent in the UK?"

_nutrients, 2021, doi:10.3390/nu13082732_

Round 1
Reviewer 1 Report
Thank you for the opportunity to read the text.
What does DPhil mean after each author? If it is not mandatory it should be removed.
In the abstract, it should be stated what method was used to compare the models.
The figures cited should be described in more detail in the Results section.
The abstract should be supplemented with information on the number of people who took part in the study.
Why do literature references follow the dot? Is this a new method and now we include literature references at the beginning of the sentence? Please explain that.
Information on strengths and limitations should be provided in a separate subsection.
In the Conclusion section you should give recommendations for the future in points.
Author Response
Reviewer 1
Thank you for the opportunity to read the text.
We thank very much Reviewer 1 for the comments on our paper.
- What does DPhil mean after each author?If it is not mandatory it should be removed.
DPhil is the title equivalent to PhD in the University of Oxford. We removed the titles as they are required by journal policy.
Page 1, line 3
Ana-Catarina Pinho-Gomes1,2,3, Asha Kaur3, Peter Scarborough3, Mike Rayner3
1 School of Population Health & Environmental Sciences, Faculty of Life Sciences & Medicine, King’s College London, London, UK
2 The George Institute for Global Health, London, UK
3 Nuffield Department of Population Health, University of Oxford, Oxford, UK
- In the abstract, it should be stated what method was used to compare the models.
We amended the abstract as follows
Page 2, line 2
A nutrient profiling model (NPM) was developed in 2005 in the UK to regulate marketing of foods to children. It was revised in 2018, but the new version has not been finalised. The Eatwell Guide (EWG) is the UK’s official food-based dietary guidelines. The aim of this study was to evaluate the agreement between the 2005 and the 2018 versions of the NPM and the EWG. Using recent National Diet and Nutrition Surveys, we estimated the healthiness of individual diets based on an EWG dietary score and a NPM dietary index. We then compared the percentage of agreement and Cohen’s kappa for each combination of the EWG score and NPM index across the range of observed values for the 2005 and 2018 versions.
- The figures cited should be described in more detail in the Results section.
We amended the Results section to describe the figures in greater detail. We also improved the figure legends to make interpretation of the figures clearer.
Page 6, line 26
Individual diets achieved zero to nine points out of ten possible points on the EWG score (Table 1). The recommendations that were met by the largest number of individuals were those related to salt, red and processed meat, and protein, whilst recommendations regarding fibre, fish, fruit and vegetables, and free sugars were the least commonly achieved (Table 2). The NPM dietary index varied between -4 and 14 for the 2005 version and between -3 and 17 for the 2018 version. The mean NPM dietary index decreased as the EWG score increased, as a lower NPM dietary index and a higher EWG score reflect a healthier diet (Table 1). For instance, for individuals who met none of the recommendations underpinning the EWG (i.e., those with an EWG score of zero), the mean NPM dietary index was 9.47 (SD 1.98) and 7.81 (SD 2.08) using the 2018 and 2005 versions, respectively. For those who achieved nine of the recommendations underpinning the EWG (i.e., those with a score of nine), the mean NPM dietary index was 1.21 (SD 2.23) and 0.30 (SD 2.47) for the 2018 and 2005 versions, respectively. For each point on the EWG score, the NPM dietary index was, in general, higher for the 2018 version than the 2005 version. This suggests that diets tended to be considered less healthy by the 2018 version than the 2005 version of the NPM dietary index at each level of the EWG score.
Diets that achieved a high EWG score had, on average, a lower NPM dietary index using both the 2005 and 2018 versions (Figures 1 and 2). The distribution of the NPM dietary index stratified by level of the EWG score also showed that diets classified as healthier according to the EWG score had a lower mean NPM dietary index using the versions from 2005 and 2018 (Figures 3 and 4). However, within each level of the EWG score, there was substantial variation in the NPM index achieved by individuals’ diets, as shown by the approximately normal distribution of the NPM index for each level of the EWG score.
There was a low, yet statistically significant, correlation between the EWG score and the NPM dietary index, with no evidence of a difference between the 2005 and 2018 versions (correlation coefficient -0.45 for the NPM version 2005 and -0.43 for the NPM version 2018, Figures S1 and S2). This showed that as the EWG score increased, reflecting diets becoming healthier, the NPM dietary index achieved by those diets decreased, which also reflects healthier diets.
Considering that adhering to six or more recommendations underpinning the EWG corresponded to a healthier diet (i.e., meeting at least half of the ten recommendations), the percentage of agreement between the EWG score and the NPM dietary index was comparable for the versions of 2018 and 2005, when an NPM dietary index threshold to define healthier diets was set at three (Table 3). When the threshold to discriminate between healthier and less healthy diets was set at a higher value, the 2018 version appeared to have better agreement with the EWG score than the 2005 version. However, Cohen’s kappa suggested that agreement between the EWG score and the NPM dietary index was low to moderate, irrespective of the NPM version. Cohen’s kappa was lower than percentage agreement because the latter did not take into account the possibility of agreement occurring due to chance. The highest kappa values were observed when a cut-off of three to five was used to consider a diet as healthy using the NPM dietary index of 2018, and two to three when the 2005 version was used. This suggests that concordance between the EWG score and the NPM dietary index is maximal when diets that are slightly healthier for the 2018 than the 2005 version.
- The abstract should be supplemented with information on the number of people who took part in the study.
We added the number of participants to the abstract. However, due to the restricted word limit, we had to make small changes to the abstract.
Page 2, line 9
A total of 3028 individual diets were assessed.
- Why do literature references follow the dot? Is this a new method and now we include literature references at the beginning of the sentence? Please explain that.
The literature references refer to the previous sentence. We amended this throughout the manuscript to avoid confusion.
- Information on strengths and limitations should be provided in a separate subsection.
We added a subheading to separate the discussion of the strengths and limitations of the study.
Page 11, line 1
Strengths and Limitations
This study has several strengths. It used actual diets from individuals living in the UK to compare the EWG score and the NPM dietary index, which are more relevant than hypothetical diets…
- In the Conclusion section you should give recommendations for the future in points.
We added a sentence to the conclusion with clear recommendations for future research.
Page 12, line 2
Further research is warranted to understand how to incorporate into FBDGs and NPMs data regarding degree of food processing, additives, and environmental considerations (e.g., carbon footprint) to allow consumers, manufacturers, retailers, and policy makers to set priorities and make informed decisions that will promote diets that are both healthy and sustainable.

Reviewer 2 Report
This is a short paper validating a nutrient profiling score for classifying healthy and unhealthy dietary patterns.
Can the authors please elaborate on the dietary assessment methods used and include information about the validation of the dietary assessment method and more details on the participants, and methods of collection.
The paper could be improved by including more explanatory information in each of the Figure and Tables legends. Can you please refer to the appropriate Figures in the results of the text, particularly for the later figures (they are not referred to in the results text at appropriate points). Also, table 4 was not provided (line 162)
Table 3: How can the Kappa and the percentage agreements differ? Please interpret the results appropriately
Line 167: Please include these results in a Figure or table. Please record the cut-points used for a healthy NPM score
Line 188: full-stop before the reference need to be corrected here and through the body of the text
Author Response
Reviewer 2
This is a short paper validating a nutrient profiling score for classifying healthy and unhealthy dietary patterns.
We thank Reviewer 2 very much for the comments on our paper.
- Can the authors please elaborate on the dietary assessment methods used and include information about the validation of the dietary assessment method and more details on the participants, and methods of collection.
We added a sentence about the methods underpinning the National Diet and Nutrition Survey and referenced the website, where in-depth information can be found.
Page 5, line 5
The NDNS rolling programme is a continuous, cross-sectional survey, which is designed to collect detailed, quantitative information on the food consumption, nutrient intake and nutritional status of the general population aged 1.5 years and over living in private households in the UK. The survey covers a representative sample of around 1000 people per year. The first stage comprises a face-to-face Computer Assisted Personal Interview with each participant, completion of an estimated four-day food diary by the participant, measurements of height and weight and collection of a spot urine sample (for those aged 4 years and over). Participants who take part in the interview and complete a food diary for at least 3 days are invited to take part in the second stage of the survey, which involves a visit from a nurse to take further physical measurements and a blood sample. A detailed description of the methods underpinning the NDNS is published elsewhere [17].
- The paper could be improved by including more explanatory information in each of the Figure and Tables legends. Can you please refer to the appropriate Figures in the results of the text, particularly for the later figures (they are not referred to in the results text at appropriate points). Also, table 4 was not provided (line 162)
We amended the Results section to explain the Figures and Tables in greater detail. We also improved the legends of the figures and tables so that they are self-explanatory and clear. We apologise for the typo regarding Table 4, which should read Table 3.
Page 6, line 26
Individual diets achieved zero to nine points out of ten possible points on the EWG score (Table 1). The recommendations that were met by the largest number of individuals were those related to salt, red and processed meat, and protein, whilst recommendations regarding fibre, fish, fruit and vegetables, and free sugars were the least commonly achieved (Table 2). The NPM dietary index varied between -4 and 14 for the 2005 version and between -3 and 17 for the 2018 version. The mean NPM dietary index decreased as the EWG score increased, as a lower NPM dietary index and a higher EWG score reflect a healthier diet (Table 1). For instance, for individuals who met none of the recommendations underpinning the EWG (i.e., those with an EWG score of zero), the mean NPM dietary index was 9.47 (SD 1.98) and 7.81 (SD 2.08) using the 2018 and 2005 versions, respectively. For those who achieved nine of the recommendations underpinning the EWG (i.e., those with a score of nine), the mean NPM dietary index was 1.21 (SD 2.23) and 0.30 (SD 2.47) for the 2018 and 2005 versions, respectively. For each point on the EWG score, the NPM dietary index was, in general, higher for the 2018 version than the 2005 version. This suggests that diets tended to be considered less healthy by the 2018 version than the 2005 version of the NPM dietary index at each level of the EWG score.
Diets that achieved a high EWG score had, on average, a lower NPM dietary index using both the 2005 and 2018 versions (Figures 1 and 2). The distribution of the NPM dietary index stratified by level of the EWG score also showed that diets classified as healthier according to the EWG score had a lower mean NPM dietary index using the versions from 2005 and 2018 (Figures 3 and 4). However, within each level of the EWG score, there was substantial variation in the NPM index achieved by individuals’ diets, as shown by the approximately normal distribution of the NPM index for each level of the EWG score.
There was a low, yet statistically significant, correlation between the EWG score and the NPM dietary index, with no evidence of a difference between the 2005 and 2018 versions (correlation coefficient -0.45 for the NPM version 2005 and -0.43 for the NPM version 2018, Figures S1 and S2). This showed that as the EWG score increased, reflecting diets becoming healthier, the NPM dietary index achieved by those diets decreased, which also reflects healthier diets.
Considering that adhering to six or more recommendations underpinning the EWG corresponded to a healthier diet (i.e., meeting at least half of the ten recommendations), the percentage of agreement between the EWG score and the NPM dietary index was comparable for the versions of 2018 and 2005, when an NPM dietary index threshold to define healthier diets was set at three (Table 3). When the threshold to discriminate between healthier and less healthy diets was set at a higher value, the 2018 version appeared to have better agreement with the EWG score than the 2005 version. However, Cohen’s kappa suggested that agreement between the EWG score and the NPM dietary index was low to moderate, irrespective of the NPM version. Cohen’s kappa was lower than percentage agreement because the latter did not take into account the possibility of agreement occurring due to chance. The highest kappa values were observed when a cut-off of three to five was used to consider a diet as healthy using the NPM dietary index of 2018, and two to three when the 2005 version was used. This suggests that concordance between the EWG score and the NPM dietary index is maximal when diets that are slightly healthier for the 2018 than the 2005 version.
- Table 3: How can the Kappa and the percentage agreements differ? Please interpret the results appropriately.
Cohen’s kappa is generally considered a more robust measure than simple percent agreement calculation, as kappa accounts for the possibility of the agreement occurring by chance. However, percentage agreement is more intuitive than kappa and hence we decided to include both measurements.
We clarified this in the manuscript as follows:
Page 7, line 27
However, Cohen’s kappa suggested that agreement between the EWG score and the NPM dietary index was low to moderate irrespective of the NPM version. Cohen’s kappa was lower than percentage agreement because the latter does not take into account the possibility of agreement occurring due to chance. This possibility increases when the population is unevenly split between the two groups, such as when the selected NPM threshold score is either very high or very low.
- Line 167: Please include these results in a Figure or table. Please record the cut-points used for a healthy NPM score.
The results for line 167 are presented in Table 3. We amended the manuscript as follows to make this clear. We also added a footnote to Table 3 to clarify that the NPM value referred to the cut-off point considered to define a diet as healthier vs less healthy.
Page 7, line 22
Considering that adhering to six or more recommendations underpinning the EWG corresponded to a healthier diet (i.e., meeting at least half of the ten recommendations), the percentage of agreement between the EWG score and the NPM dietary index was comparable for the versions of 2018 and 2005, when an NPM dietary index threshold to define healthier diets was set at three (Table 3). When the threshold to discriminate between healthier and less healthy diets was set at a higher value, the 2018 version appeared to have better agreement with the EWG score than the 2005 version.
- Line 188: full-stop before the reference need to be corrected here and through the body of the text.
We corrected this throughout the manuscript.

Round 2
Reviewer 1 Report
I am glad that the authors introduced the suggested changes and explained the unclear issues.
I have no further comments.